


# A process-based *Sphagnum* plant-functional-type model for implementation in the TRIFFID Dynamic Global Vegetation Model

Richard Coppell[1], Emanuel Gloor[1], Joseph Holden[1]

[1]water@leeds, School of Geography, University of Leeds, Leeds, LS2 9JT, UK

*Correspondence to*: Richard Coppell (r.coppell@leeds.ac.uk)

**Abstract.** Peatlands are important carbon stores and *Sphagnum* moss represents a critical peatland genus contributing to carbon exchange and storage. However, gas fluxes in *Sphagnum*-dominated systems are poorly represented in Dynamic Global Vegetation Models (DGVMs) which simulate, via incorporation of Plant Functional Types (PFTs), biogeochemical and energy fluxes between vegetation, the land surface and the atmosphere. Mechanisms characterised by PFTs within DGVMs include
photosynthesis, respiration and competition and, in more recent DGVMs, sub-daily gas-exchange processes regulated by leaf stomata. However, *Sphagnum*, like all mosses, are non-vascular plants and do not exhibit stomatal regulation. In order to achieve a level of process detail consistent with existing vascular vegetation PFTs within DGVMs, this paper describes a new process-based non-vascular-PFT model that is implemented within the TRIFFID DGVM used by the JULES land surface model. The new PFT model was tested against extant published field and laboratory studies of peat assemblage-net primary
productivity, assemblage-gross primary productivity, assemblage respiration, water-table position, incoming photosynthetically active radiation, temperature, and canopy dark respiration. The PFT model's parameters were roughly tuned and the PFT model easily produced curves of the correct shape for peat assemblage-net primary productivity against water-table position, incoming photosynthetically active radiation and temperature, suggesting that it replicates the internal productivity mechanism of *Sphagnum* for the first time. Minor modifications should also allow it to be used across a range of
other bryophytes enabling this non-vascular PFT model to have enhanced functionality.

## 1 Introduction

Around 600Gt of carbon exists in peat that has accumulated in temperate and cold biomes since the Last Glacial Maximum (Yu et al. 2010, Yu 2011). This is about 25% of the carbon stored in deep soils (Jobaggy & Jackson 2000). Peat accumulates in areas where precipitation exceeds evapotranspiration and there is impeded drainage due to the landscape configuration or
underlying impermeable layers. There is uncertainty about feedback mechanisms between peatland systems and future climate change. For example, warming may enhance decomposition rates of peat and thus release of carbon to the atmosphere, but it may also enhance net primary production (NPP) and thus mitigate anthropogenic greenhouse warming (Loisel et al. 2012; Charman et al. 2013; O'Connor et al. 2010). This means that the net result on greenhouse gas fluxes regionally and globally is not clear and there is therefore a need for improved climate-land surface feedback models that incorporate such peatland
processes (Limpens et al. 2008; Frolking et al. 2009; Hayman et al. 2014).



Dynamic Global Vegetation Models (DGVMs) simulate carbon, water and energy fluxes between vegetation, the atmosphere and the land surface (e.g. Clark et al. 2011; Haxeltine & Prentice, 1996; Sitch et al. 2003). DGVMs may be run independently with prescribed climate or they can be coupled to climate models. For example, the TRIFFID DGVM is a sub-model of the

JULES land surface model which is in turn part of the Hadley Centre climate model (Clark et al. 2011; Pope et al. 2000). As such, DGVMs could be used in understanding past and future change in peatlands and their interactions with global climate. DGVMs simulate the effects of a limited number of vegetation classes (rather than every single species) in each horizontal grid square of the model, reflecting the assumption that regional vegetation assemblages are dominated by a small number of Plant Functional Types (PFTs) (Brovkin et al. 1997; Haxeltine & Prentice, 1996). The main vegetation processes represented

in DGVMs include photosynthesis and accumulation of atmospheric carbon in plant tissues, respiration, the accumulation of litter from dead plant material and plant competition. Therefore, for use in peatlands, DGVMs require appropriate incorporation of dominant peatland PFTs, and their associated relevant carbon exchange and litter accumulation processes.

In the current generation of DGVMs, vegetation-atmosphere $CO_2$ exchange is commonly represented at sub-daily time-steps,

incorporating regulation of gas exchange in vascular plants by leaf stomata that close in response to moisture deficit in the leaf (e.g. Cox, et al. 1998; Clark et al. 2011; Gao et al. 2013). *Sphagnum* mosses are a dominant genus within peatlands. At least half of the carbon in northern peatlands is dead matter from *Sphagnum* (Clymo & Hayward, 1982). However, there is a key problem with incorporating *Sphagnum* into DGVMs: *Sphagnum* is a non-vascular plant that lacks stomatal gas-exchange. Therefore, *Sphagnum* mosses show quite different responses of photosynthesis and respiration to changes in their immediate

environment compared to vascular plants and this means they cannot be simply incorporated into current DGVMs. For example, Riutta et al. (2007) examined gross primary production (GPP) of *Sphagnum* compared to vascular wetland plants, for different environmental conditions. Unlike vascular wetland plants, *Sphagnum* GPP showed a high tolerance to waterlogging, and little short-term temperature-dependence with much higher rates of production for temperatures below 20°C than vascular wetland plants.

*Sphagnum* also causes changes in the local physical and chemical environment, slowing the rate of decay (Rydin et al 2006). *Sphagnum* cell walls and litter contain insoluble polysaccharides that inhibit microbial mineralisation, while the release of uronic acids from *Sphagnum* acidifies the soil water and hinders the decay of adjacent litter from other plants (Hájek et al. 2010; Van Breemen 1995). Thus, *Sphagnum* is associated with high rates of plant litter accumulation in cool, wet areas

resulting in peatland extents and depths that are larger than they would be without *Sphagnum* (Rydin & Jeglum, 2006).

Given the importance of *Sphagnum* to peatland systems and the global carbon store, it is essential that *Sphagnum* is appropriately incorporated into the latest-generation DGVMs. Variations do exist in *Sphagnum* assemblage behaviour – for example, wetland sub-environments such as hummocks, hollows and pools have different assemblages of *Sphagnum* species



although similarities also exist (Rydin 1986). However, DGVMs also require the number of PFTs to be limited, and previous studies (Druel et al. 2017; Qiu et al. 2018; Wania 2009a, 2009b; Yurova et al. 2007) have found improved fit of peatland ecosystem NPP with field $CO_2$ exchange data using very simple *Sphagnum* PFTs. Therefore, we sought to develop, as a first attempt, a single process-based *Sphagnum* PFT that has a higher level of detail than these previous studies in a current-

generation DGVM, having a consistent level of detail with existing vascular PFTs in the same DGVM. We chose the TRIFFID DGVM because it has very widespread use as part of JULES and the Hadley Centre Climate Model and it has been validated across many biomes (e.g. Clark & Gedney, 2008; Cox et al 1998; Cox et al 2001; Hughes et al 2004; Harper et al 2018). Despite its widespread use, however, a major advancement is needed in the functionality of the model by improving the representation of the peatland biome. This is because it does not have any module that reproduces the vegetation or hydrology

mechanisms of peatlands (e.g. see Harper et al's (2018) description of the vegetation types represented in a very recent version of JULES).

Other DGVMs that have simple *Sphagnum* PFTs incorporated are coupled to a peat-soil-model, and were validated as whole ecosystems not as individual PFTs. A *Sphagnum* plant respiration and production model was developed by Yurova et al (2007)

as a modification of the Lund-Potsdam-Jena General Ecosystem Simulator (LPJ-GUESS). The LPJ base-model is simpler than TRIFFID, lacking detail such as leaf-gas-conduction and therefore not making any model-code-level distinction between vascular and non-vascular plants. Therefore, LPJ's *Sphagnum* PFT just has different parameters without changing the base-code of the model. The ORCHIDEE-PEAT model (Qiu et al. 2018) uses its existing C3 grass PFT with stomatal gas-regulation to simulate peatland NPP, calibrating it at each site using locally measured $V_{max}$ values (a key enzyme-controlled reaction rate

in photosynthesis dependent on the local environment as well as inherited properties of the plants). This local calibration against $V_{max}$ is recognised by the authors as a major adjustment factor to cover multiple uncertain parameters, adjusting for the fact that non-vascular properties of some of the vegetation in the test-sites are not accommodated in the model code. Similarly, Druel et al (2018) in another version of ORCHIDEE used its existing C3 grass vascular model and parameters to simulate non-vascular plants (including all mosses as well as *Sphagnum* in a single PFT), modifying the parameters to minimise the effects

of stomatal regulation of leaf-gas exchange from the base vascular model, but short of changing the base model itself to resemble the non-vascular function of *Sphagnum*. These ORCHIDEE and LPJ sub-models represented whole peat-ecosystems including soil-processes and were validated against $CO_2$ -eddy-covariance measurements; they were not validated at the level of isolated individual PFTs. In this study, we have focussed on specific *Sphagnum*-processes instead of trying to approximate *Sphagnum*'s behaviour without changing the base model. This is to try to replicate *Sphagnum*'s production and respiration

mechanism in more detail than other DGVMs have done so far, at this isolated PFT level. For the first time with any non-vascular PFT as far as we are aware, this model should permit direct comparison with organism-level productivity and respiration measurements, permitting the first attempt at a general model for *Sphagnum* photosynthesis and respiration at this higher level of detail. Since this work does not include the development of a coupled peat-soil model then the model-output





cannot be validated against $CO_2$ eddy-covariance measurements (such as used by Qiu et al 2018) since we are not simulating whole-ecosystem NPP.

An increased level of functional detail is introduced in section 2 where we outline the key biological and physical traits of
*Sphagnum* that require new functions in TRIFFID, with an explanation of how we have incorporated specific *Sphagnum* traits within TRIFFID. We present the outputs of model runs in section 3. We test the model with available data from recently published studies, which measured assemblage-NPP, assemblage-GPP, assemblage respiration, WTP (water-table position), PAR (photosynthetically active radiation), temperature, canopy dark respiration and $V_{max}$. Canopy dark respiration is the respiration in leaves that occurs in cell mitochondria as opposed to photosynthesising cell components, and $V_{max}$ is the
maximum rate of carboxylation of rubisco, which is a key photosynthesising enzyme that controls primary production. Section 4 concludes by outlining further potential developments of the new *Sphagnum* PFT model and its wider use.

## 2. Model development and theory

*Sphagnum* exchanges gas directly through cell walls. Individual *Sphagnum* stems and leaves are spongy and porous, and they grow in tightly packed assemblages. The surface shoots of these *Sphagnum* assemblages have been observed to retain a lot of
water in this way, having up to ten times as much water by mass as the dry-mass of the plants themselves (Hayward & Clymo 1982; Rydin 1985; Rydin & Jeglum 2006; Strack et al. 2009). This water is held and sucked up by capillarity in the surface *Sphagnum* matter and deeper *Sphagnum* litter (e.g. Charman 2002; Rydin & Jeglum 2006; Van Breemen 1995). These characteristics mean that *Sphagnum* plants have clear differences to vascular plants in terms of physical shape and the processes that occur within them. Where these characteristics are relevant to simulating photosynthesis and plant respiration ($R_p$) in a
PFT model, they are described in Table 1 in comparison to a pre-existing example of a vascular PFT that has been developed and tested in TRIFFID. Table 1, therefore, illustrates the basis for modifications that are required to incorporate *Sphagnum* within the TRIFFID DGVM.



**Table 1. A comparison of *Sphagnum* processes with those of typical C3 vascular plants**

| Process | C3 Graminoids | *Sphagnum* |
|---|---|---|
| **Life form** | Able to survive as individuals across wide range of soil conditions, although few survive long periods total saturation. Consist of roots (below ground) and the shoot that has stems, leaves and seed-heads. Often exist in large assemblages with other grasses and other vascular plants. | Only ever occur in assemblages. Only exist where the soil normally has high moisture content and some species tolerate long periods of total saturation. Consist of closely packed branching stems. Along the stems are 'stem leaves' and the whole assemblage forms 'mats' where only the top few cm (the capitulum) receive enough light to photosynthesise. |
| **Leaf anatomy** | Several cell layers, water impermeable cuticula, punctuated with stomata. | Leaf is single cell layer containing photosynthesising and non-photosynthesising cells with no stomata. |
| **Leaf-air gas exchange** | Regulated by stomata that open fully or partially to specifically allow transpiration, but also conserve moisture by closing up under water-stress of the leaf or when there are strong humidity gradients. | Constant gas exchange via molecular diffusion between photosynthesising cells and either directly with atmosphere or via thin water-film between leaf and atmosphere. |
| **Photosynthesis** | C3 pathway | C3 pathway |
| **Water uptake and transport** | Xylem vessels connect the roots vertically to the leaves. Strongly negative water potential at leaf surface is due to transpiration from the leaf. Water drawn up under strong hydraulic gradients from roots. Waxy cuticula leaf surface prevents direct leaf-water uptake. | Water uptake and release from much of the plant surface. Vertical water transport occurs up the exterior of the stem, including deeper dead stem material, and underlying peat by capillary action. No clear dividing line between these formations. Assemblages also intercept and absorb water from direct precipitation. |
| **Water storage and stress** | Not desiccation tolerant. Metabolism might slow under stress but needs to be maintained. Water stored in living cells; high internal water storage is a requirement for plant structure and survival. Stomata close during drought to limit transpiration but high soil water tension leads to the plant being unable to access soil water, resulting in cell wall collapse and mortality. | Desiccation tolerant. Metabolism suspended until wet periods return. Quick to recover after drought. Water storage in spongy assemblages. The leaves consist of small living chlorophyllose cells and large structural dead hyaline cells that store lots of water. Closed canopy shelters underlying peat from direct sunlight. Peat typically has very high moisture content even after drought. |





## 2.1 Life form

*Sphagnum* assemblages exist at a spatial scale of up to $100\,km^2$ (e.g. Baird et al. 2009; Bragg & Lindsay 2003; Rydin & Jeglum 2006). *Sphagnum* assemblages are closely packed 'mats' within which individuals, of the order of 10 cm in length, are vertically oriented. They are typically damp or saturated in their lower extents, and their leaves do not exhibit significant
internal water transport and have no vascular system (Rydin et al. 2006, Rydin & Jeglum, 2006; Van Breemen 1995). Due to the lateral packing of the assemblages, only the top few centimetres of *Sphagnum* mats receive enough light to photosynthesise (Rydin & Jeglum, 2006). This photosynthesising surface section is called the capitulum. Different *Sphagnum* species show varying degrees of environmental tolerances and responses, and favour particular microenvironments of topography and wetness. Nevertheless, *Sphagnum* species show similar *forms* of reactions to variations in environmental drivers (Johnson et
al. 2015).

## 2.2 Leaf anatomy and air-gas exchange

The one-cell-thick *Sphagnum* leaf has green photosynthesising cells that alternate with larger hyaline cells. The hyaline cells are rigid, transparent, porous, structurally support the plant, and store water through capillarity. The cells die when fully
developed within the living plant, but they retain their structural and hydrological functions (Rydin et al. 2006, Rydin & Jeglum 2006). There is no active control of the rate of the conductance of any substance between the photosynthesising cells in *Sphagnum* and the atmosphere (Rydin & Jeglum 2006). These photosynthesising cells are directly in contact with the atmosphere or submerged under water. This is different to vascular plants, which have an opening and closing apertures in the leaf surface that permit atmospheric exchange (e.g. Amthor & Baldocchi, 2001).

## 2.3 Photosynthesis

There are several different sets of processes within the photosynthesising cells of different types of plants, but each plant type exhibits only one such carbon fixation pathway. The most common is the C3 carbon fixation pathway, which *Sphagnum* also
shows (Loisel, et al. 2009; Price, et al. 1997). $R_p$ is equal to the carbon lost to the atmosphere because of energy use by the plant including respiration from leaves, stems and roots. GPP is equal to the carbon that is fixed by photosynthesis as carbohydrate. NPP is the arithmetic difference between GPP and $R_p$, which results in the physical growth of the plant through the net assimilation of carbon (Amthor & Baldocchi, 2001; Begon, et al. 2005).





## 2.4 Water transport, storage and stress

*Sphagnum* plants store water within densely packed stems and leaves, incorporating hyaline cells. The volume stored varies between species (Rydin & Jeglum 2006). Capillarity enables vertical water transport to occur up the exterior of the stem from the underlying peat, including along deeper dead *Sphagnum* material (Hayward and Clymo 1982; Rydin & Jeglum 2006; Van

Breemen 1995). There is very little internal vertical water transport within the stems. The assemblages also intercept and absorb water from direct precipitation (Hayward and Clymo 1982; Rydin & Jeglum 2006; Thompson & Waddington 2008; Van Breemen 1995), while the closed canopy shelters underlying peat from direct sunlight (Bu et al. 2013; Charman 2002). *Sphagnum* is desiccation-tolerant, and it suspends its metabolism until wet periods return after which it is quick to recover after drought (Hájek & Beckett 2008).

## 2.5 Model components

Given the above information, we take as a template the carbon exchange equations of TRIFFID and modify them according to specific characteristics of *Sphagnum*. There are six processes and mathematical process descriptions, in Tables 1 & 2. The new model does not explicitly accommodate water storage, transpiration, desiccation-tolerance or variable *Sphagnum* canopy

$CO_2$ uptake and release related to microform position (Wania et al. 2009b). These would require new process understanding and datasets and so they have not been addressed in this version of the TRIFFID *Sphagnum* PFT model. However, there are 11 important changes to the parameters of the model. Seven parameters were developed using published field and laboratory studies (Table 3), and four used as shaping parameters (Table 4). The following paragraphs describe this parameter development.

The new PFT model simulates photosynthesis of a *Sphagnum* assemblage, which is not simply scaled up individual-behaviour. This is shown in a new empirical water-uptake equation (the right-hand-column for point 1 in Table 2) which simulates the assemblage-water-uptake mechanism (as described in section 2.4). It is based on a similar mechanism for vertically distributed vascular roots in existing TRIFFID vascular PFTs (Cox et al 1998). Cox et al. (1998) found that this has the advantage of

programmatic simplicity, employing fewer variables than explicit simulation of water-uptake, whilst still giving a good model fit. This approach directly utilises the unitless volumetric water content of the photosynthesising *Sphagnum* capitulum as the desiccation-stress factor. The specific desiccation stress factor for the new *Sphagnum* PFT model was based on field observations derived from Strack et al. (2009) who plotted an empirical function of capitulum water content against the water-table position (WTP) for *Sphagnum* assemblages and who also found a strong linear relationship between surface volumetric

moisture content and assemblage production. Murray et al. (1989) and Riutta et al. (2007), studying bog ecosystems in the foothills of the Philip Smith Mountains, Alaska, and the Lakkasuo bog in southern Finland, respectively, recognised that the maximum *Sphagnum* photosynthetic production value occurs when the water table is just below the ground surface. A fixed



maximum value was therefore used to reproduce this, when the WTP is less than 10 cm below the ground surface, incorporated as a limit in the *Sphagnum* equation shown on the right-hand-column for point 1 in Table 2. This new *Sphagnum* equation provides an overall non-vascular desiccation-stress function, which replaces Cox et al's (1998) vascular desiccation-stress function on the left column of point 1 in Table 2. For leaf air-gas exchange, where a simple scaled-up 'big leaf' approach is

used, a fixed value for leaf conductance has been applied in the new *Sphagnum* PFT model, replacing the variable stomatal conductance used in the TRIFFID vascular PFT model (point 2 in Table 2). This approach is in place of Williams & Flanagan's (1998) explicit simulation of varying *Sphagnum* leaf conductance, abstracting away from their higher level of detail in favour of reduced model-complexity (for example dead *Sphagnum* matter capillarity at the top of the soil column is not incorporated in any DGVMs yet). This follows similar successful approaches in Cox et al. (1998) for vascular plants, and Wania et al.

(2009b) and Yurova et al (2007) for *Sphagnum* growth in the LPJ model.

The equations that simulate photosynthesis and respiration processes that occur within cells, which are unchanged by assemblage-behaviour, are in points 3 and 4 in Table 2. To simulate the photosynthesis biochemical pathway in the new *Sphagnum* model, the existing C3 carbon fixation function within the TRIFFID vascular PFT sub-model was used to form the

basic model. GPP for vascular PFTs is simulated in TRIFFID, as in many other DGVMs, as a function of the minimum of three limiting rates to fix carbon within the photosynthesising cells of a leaf (Clark et al. 2011). These are the light limited rate $GPP_l$, the Rubisco-limited rate, $GPP_r$, and the rate of transport of photosynthetic products by the plant, $GPP_e$, which are simulated using the equations on the left-hand column for point 3 in Table 2. In the new *Sphagnum* model, the conductance between the atmosphere and *Sphagnum's* photosynthesising tissues is a fixed value representing permanent exposure of the

tissues to the atmosphere, resulting in a similar approach to Druel et al. (2017) and Dimitrov et al. (2011) to simulating non-vascular plants' leaf-conductance. The solutions for internal leaf partial pressure ($c_{ri}$ and $c_{li}$ respectively) as functions of the photosynthesising processes in the leaf and atmospheric pressure are then solutions of the quadratic equation given on the right-hand column for point 3 in Table 2, thus giving solutions for $GPP_r$ and $GPP_l$ by substitution. A smoothed minimum value to simulate overall GPP (using the existing TRIFFID smoothing function in simple form on the left-hand column in point 3 of

Table 2, and in detailed form in Appendix A) is then calculated from these two values (and the unaltered photosynthetic product transport-limited GPPe). The plant respiration equations (point 4 in Table 2) are unchanged, but the parameters saw significant changes, see sub-section 2.6.

The gas assimilation-inflow continuity equation and surface energy balance equation (respectively points 5 and 6 in Table 2)

are unchanged as these describe simple physics that should be the same between non-vascular *Sphagnum* and vascular plants.

The new *Sphagnum* PFT model requires changed parameters because *Sphagnum* has different physical dimensions to the other PFTs already represented in TRIFFID. As a result, there were 11 changes to PFT parameters needed in the new TRIFFID *Sphagnum* PFT model (Tables 3 and 4). Of the physiology parameters (Table 3), three are parameters for *Sphagnum* respiration





taken from observation-based parameters tested in LPJ-GUESS (N10, $T_{upp}$, $T_{low}$) while four have had their values established from existing field and laboratory literature on *Sphagnum* (tleaf_of, infil_f, catch0, Canht_ft). We could not find precise data to constrain the parameters LAI, fd and rg (Table 4) so they were used to shape the output respiration function to the output field data. For example, Yurova et al. (2007) and Bond-Lamberty & Gower (2007) give rough ranges for these parameters.

5    Appendix B provides the detailed steps taken to set the shaping parameters, which resulted in an initial calibrated model whose outputs are in section 3.





**Table 2. Processes and mathematical descriptions of carbon exchange plant growth in C3 PFTs in TRIFFID and the modifications made to them according to the specifics of *Sphagnum* for the new *Sphagnum* PFT model.**

| TRIFFID – C3 vascular plant | *Sphagnum (see Appendix A for derivations)* |
|---|---|
| **(1) Carbon gain of plant – mass balance** $$A = (GPP - R_d)\beta$$ $$\beta_{vascular} = \sum \begin{cases} 1, & \theta > \theta_c \\ \dfrac{\theta - \theta_w}{\theta_c - \theta_w}, & \theta_w < \theta \le \theta_c \\ 0, & \theta > \theta_w \end{cases}$$ $\beta_{vascular}$ is summed for all the soil layers underlying the PFT | Has the following changes: $$\beta_{Sphagnum} = \begin{cases} 1, & WTP > -0.1m \\ e^{\left(\frac{0.1 + WTP}{0.16}\right)}, & WTP \le -0.1m \end{cases}$$ |
| **(2) Stomatal conductance** $$g_s = \frac{\alpha A}{(c_{atm} - \Gamma)\left(1 + \dfrac{D}{D_0}\right)}$$ 'Simplified Leuning model' - the Leuning (1995) model of stomatal conductance in a vascular plant, was simplified by Cox et al. (1998) by setting optimal minimum canopy conductance to zero. | Has the following changes: $$g_s = 1.6 * 0.0237 * 0.07 \ ms^{-1}$$ |
| **(3) Photosynthesis machine** $$GPP \approx min(GPP_r, GPP_l, GPP_e)$$ $$GPP_r = V_m\left(\frac{c_i - \Gamma}{c_i + K_c(1 + O_a/K_0)}\right)$$ $$GPP_l = 0.08(1 - \omega)I_{par}\left(\frac{c_i - \Gamma}{c_i + 2\Gamma}\right)$$ $$GPP_e = 0.5V_m$$ $$\Gamma = \frac{O_a}{2\tau}$$ $$\tau = 2600 f_T(0.57)$$ | Has the following changes: $$GPP_r = V_m\left(\frac{c_{ri} - \Gamma}{c_{ri} + K_c(1 + O_a/K_0)}\right)$$ $$GPP_l = 0.08(1 - \omega)I_{par}\left(\frac{c_{li} - \Gamma}{c_{li} + 2\Gamma}\right)$$ $$c_{ri}, c_{li} = \frac{-BB + \sqrt{BB^2 - 4AACC}}{2AA}$$ $$AA_r = \frac{G}{V_m} \ , \ AA_l = \frac{G}{I} \quad BB_r = \frac{G}{V_m}(K - c_{atm}) - R_d + 1$$ |





$$f_T(T_c) = q_{10\_leaf}^{0.1(T_c-25)}$$

$$BB_l = \frac{G}{I}(2\Gamma - c_{atm}) - R_d + 1$$

$$CC_r = -\left(\frac{G}{V_m}Kc_{atm} + KR_d + \Gamma\right)$$

$$CC_l = -\left(\frac{G}{I}2\Gamma c_{atm} + 2\Gamma R_d + \Gamma\right)$$

where the following simplifying terms are used:

$$G = \frac{g_s}{1.6\bar{R}T^*\beta}, \quad I = 0.08(1-\omega)I_{par}, \quad K = K_c\left(1+\frac{0_a}{K_0}\right)$$

**(4) Respiration**

$$R_p = r_g(\Pi_G - R_{pm}) + R_{pm}$$

(plant respiration = growth respiration + maintenance respiration)

$$R_{pm} = 0.012 \cdot R_{dc}\left(\beta + \frac{N_r + N_s}{N_l}\right)$$

(maintenance respiration)

$$R_d = f_d \cdot V_m$$
$$R_{dc} = \sum(R_d)$$

(canopy dark respiration)

Equations unchanged, new parameters taken from tested *Sphagnum* parameters in LPJ-GUESS/WHY which in turn were originally based on field data (see appendix B).

**(5) Continuity equation**

Assimilation = inflow

$$A = \frac{g_s}{1.6\bar{R}T_*}(c_{atm} - c_i)$$

Unchanged

**(6) Surface energy balance**

$$R_S = SW_N + LW_\downarrow - \sigma T*^4 - H - LE - G_0$$

Unchanged

Parameter names in this table:

A - net leaf photosynthesis (mol $CO_2$ m$^{-2}$s$^{-1}$)





AA, BB, CC – quadratic equation parameters

$AA_r$, $BB_r$, $CC_r$ - quadratic equation parameters for case of Rubisco-limitation of GPP

$AA_l$, $BB_l$, $CC_l$ - quadratic equation parameters for case of light-limitation of GPP

$c_{atm}$ - atmospheric $CO_2$ partial pressure (Pa)

$c_i$ - internal leaf $CO_2$ partial pressure (Pa)

$c_{ri}$ – Rubisco-limited internal leaf $CO_2$ partial pressure (Pa)

$c_{li}$ – light-limited internal leaf $CO_2$ partial pressure (Pa)

D - Humidity deficit at the leaf surface (g kg$^{-1}$)

$D_0$ - Leaf surface humidity deficit parameter for the 'Leuning' closure (g kg$^{-1}$)

$f_d$ - the dark respiration scaling factor

fT - the standard Q10 temperature dependence

$g_s$ – leaf surface conductance (stomatal conductance in vascular plants) (ms$^{-1}$)

G, I, K – terms to simplify the layout of equations in the *Sphagnum* photosynthesis machine

GPP – leaf Gross Primary Production (mol $CO_2$ m$^{-2}$s$^{-1}$)

$GPP_e$ – GPP rate limited by transport of photosynthetic products by the plant (mol $CO_2$ m$^{-2}$s$^{-1}$)

$GPP_l$ – light limited rate of GPP (mol $CO_2$ m$^{-2}$s$^{-1}$)

$GPP_r$ – Rubisco-limited rate of GPP (mol $CO_2$ m$^{-2}$s$^{-1}$)

$G_0$ – heat flux into the ground (see Best et al. (2011) for detail inside this term) (Wm$^{-2}$)

H – sensible heat flux (Wm$^{-2}$)

$I_{par}$ - incident photosynthetically active radiation (mol photons m$^{-2}$ s$^{-1}$)

$K_c$, $K_0$ - Michaelis Menten parameters for $CO_2$ and $O_2$ (Pa) (see Clark et al. (2011))

LE – heat flux due to evapotranspiration (latent heat of vaporisation, L x moisture flux, E) (Wm$^{-2}$)

$LW_↓$ - incoming longwave radiation (Wm$^{-2}$)

$N_l$ - nitrogen content of leaf portion of plant (mol $CO_2$ m$^{-2}$ s$^{-1}$)

$N_r$ - nitrogen content of root portion of plant (mol $CO_2$ m$^{-2}$ s$^{-1}$)

$N_s$ - nitrogen content of stem portion of plant (mol $CO_2$ m$^{-2}$ s$^{-1}$)

$O_a$ - partial pressure of atmospheric oxygen (Pa)





Q10_leaf – Q10 temperature dependence coefficient

$r_g$ – growth respiration coefficient

$R_{pm}$ - plant maintenance respiration (kgC $m^{-2}s^{-1}$)

$R_d$ - leaf dark respiration (mol C $m^{-2}s^{-1}$)

$R_p$ – plant respiration (kg C $m^{-2}s^{-1}$)

$R_S$ – surface energy balance ($Wm^{-2}$)

$R_{dc}$ - canopy dark respiration (mol C $m^{-2}s^{-1}$)

$SW_N$ – balance of shortwave radiation ($Wm^{-2}$)

$T_c$ - canopy (leaf) temperature (°C)

$T_*$ - the leaf surface temperature (K)

$V_m$ – temperature-dependant maximum rate of carboxylation of the Rubisco enzyme ($CO_2$ $m^{-2}$ $s^{-1}$)

WTP – water table position (m)

$\alpha$ - Quantum efficiency [mol $CO_2$ (mol PAR, photons)$^{-1}$]

$\beta$ - desiccation stress factor, unitless ($0 \leq \beta \leq 1$)

$\Pi_G$ – PFT-canopy GPP rate (kgC $m^{-2}s^{-1}$)

$\theta$ - mean soil water concentration in the root zone

$\theta_c$, $\theta_w$ - vascular PFT-specific critical and wilting soil moisture concentrations respectively

$\sigma$ - Stefan Boltzmann constant ($Wm^{-2}$ $K^{-4}$)

$\tau$ - Rubisco specificity for $CO_2$ relative to $O_2$

$\Gamma$ - $CO_2$ photorespiration compensation point (Pa)

$\omega$ - the leaf scattering coefficient for PAR



**Table 3. TRIFFID PFT constrained physiology parameters changed for the new *Sphagnum* PFT model.**

| Function | Symbol | Value | Description / references |
|---|---|---|---|
| **Canopy Height** | Canht_ft | 0.03m | Height of leaves above ground (Smolders, et al. 2001) |
| **Minimum canopy capacity** | catch0 | 0.25 kg m$^{-2}$ | Bond-Lamberty & Gower (2007) |
| **Infiltration enhancement factor.** | infil_f | 1.0 | Reduced compared to grasses (2.0) reflecting lower hydraulic conductivity |
| **Top leaf nitrogen concentration** | Nl0 | 0.02 kg N/kg C | Wania, et al. (2009b) (Wania, 2007), in approximate agreement with (Rice 2000) and Aerts et al (2009) |
| **Temperature below which leaves are dropped** | tleaf_of | 0 Kelvin | Representing no lower limit |
| **Lower temperature for photosynthesis** | Tlow | 5.0 Celsius | Gerdol, et al. (1998); Wania, et al. (2009b) |
| **Upper temperature for photosynthesis** | Tupp | 36.0 Celsius | Gerdol, et al. (1998); Wania, et al. (2009b) |





**Table 4. TRIFFID PFT physiology shaping parameters for the new *Sphagnum* PFT model, compared with equivalent parameters in LPJ-GUESS *Sphagnum* PFT.**

| Function | Symbol | Value | |
|---|---|---|---|
| | | LPJ-GUESS | TRIFFID |
| **Dark Respiration Scaling Factor** | $f_d$ | 0.03 | 0.03 |
| **Growth Respiration Coefficient** | $r_g$ | 0.5 | 0.1 |
| **Non-Photosynthetic Tissue Respiration** | $r_m$ | 2.0 | Not used |
| **Leaf Area Index** | LAI | Not used | 3, hummock/bog species |
| | | | 1, lawn/fen species |





## 2.6 Model Calibration

**Table 5. Literature sources of quantified *Sphagnum* respiration and photosynthesis functions.**

| Reference | Model Functions Adopted in the new *Sphagnum* PFT model<br>*[Functions used for output comparisons]* | *Sphagnum* species quantified | Season when measurements taken |
|---|---|---|---|
| **Aerts et al (2009)** | Leaf nitrogen concentration | *S Fuscum* | Summer |
| **Hayward & Clymo (1982)** | Capitulum volumetric water content against WTP, similar shaped function to Strack et al. (2009) (below) | *S. Capillifolium*<br>*S. Papillosum* | Not given |
| **Murray et al. (1989)** | Photosynthetic maximum at near surface water table, agrees with Riutta et al.(2007) (below) | *synthesised studies of many arctic species* | Summer |
| **Rice (2000)** | Leaf nitrogen concentration | *S. Recervum*<br>*S. Palustre*<br>*S. Tenerum* | Spring and summer |
| **Riutta et al. (2007)** | Photosynthetic maximum at near surface water table, but photosynthetic and respiration responses are otherwise approximately flat against WTP<br><br>*[NPP against WTP, temperature and PAR]* | *S. Papillosum*<br>*S. Fallax*<br>*S. Flexuosum* | 'Growing season' |
| **Strack et al. (2009)** | Capitulum volumetric water content against water table position, after first order correction for presence of vascular vegetation<br><br>*[NPP against WTP]* | *S. Rubellum* | Very late spring, summer |
| **Williams & Flanagan (1998)** | Maximum leaf $CO_2$ conductance<br><br>*[Cell dark respiration and Vm]* | *S. Rubellum*<br>*S. Tenellum*<br>*S. Capillifolium*<br>*S. Fuscum* | Summer |





We could only find one set of data that was useful for general model calibration (Table 5 – Strack et al. 2009) and two for validation (Table 5 – Riutta et al. 2007; Williams & Flanagan 1998). The remaining data sets listed in Table 5 either gave very similar data to these three sources, or individual parameters for the model. Strack et al (2009) presented GPP and $R_p$ field-data

from a *Sphagnum*-dominated bog ecosystem (nutrient-poor peatland with a shallow to deep water-table) in late spring and summer. Having GPP and $R_p$ disaggregated was very useful for calibrating our detailed *Sphagnum* model, and as far as we are aware, these data are the only such published data. Strack et al (2009) studied a mixed ecosystem with at least 90% *Sphagnum* cover, so, unfortunately, the field-data did not represent isolated *Sphagnum* behaviour. However, we found a first-order correction, based on data from Riutta et al (2007), who showed that peatland vascular vegetation NPP (which is GPP-$R_p$)

against WTP is approximately horizontal, or uncorrelated. Additionally, Riutta's *Sphagnum* GPP data asymptotically approaches zero for deep water tables, which leads to desiccation of the capitulum, a theory supported by observations from Bewley et al. (1978), Hájek & Beckett (2008) and Hayward & Clymo (1982). Therefore, a first order correction was applied by simply subtracting a fixed positive offset (assumed to represent the influence of vascular plants) from the $R_p$ and GPP data of Strack et al (2009) so it has $R_p$ and GPP approaching zero for very low WTP values. The resultant NPP data was in this way

transformed to give the calibration data shown in Figure 1.

Our model was calibrated to reproduce the transformed GPP, NPP and $R_p$ data of Strack et al (2009). We used fixed environmental model-inputs (PAR, temperature and WTP), running the model like a controlled laboratory without varying climate inputs. $r_g$ (growth respiration coefficient – see Table 4) was adjusted to set the $R_p$ curve in the correct position. The

NPP curve merely is the algebraic sum of these two curves. This parameter-fitting exercise is described fully in Appendix B. The outcomes are in Figure 1.



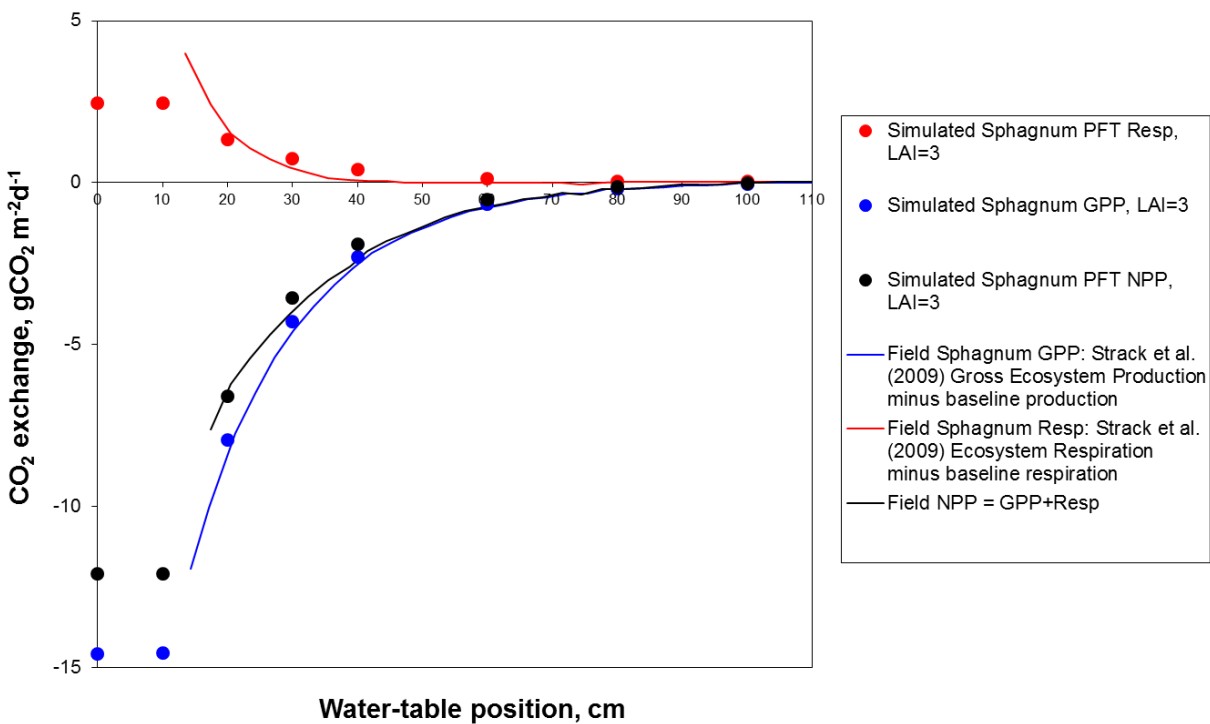

**Figure 1. Model outputs of respiration, GPP, and NPP against WTP, calibrated with field measurements from a *Sphagnum*-dominated peatland constructed using data derived from Strack et al (2009).**

## 3. Results

The calibrated *Sphagnum* PFT model was tested against the limited data available from Riutta et al (2007) and Williams & Flanagan (1998) and gave good visual matches against these quite complex measured functions of varying ecosystem productivity and respiration. The degree of fit was not measured quantitatively, because getting the shape of all the various output functions of the model right was a much higher priority than the accuracy of fit, and the test-data is very limited. The model reproduced the shape of the seven different curves in both calibration and test datasets with a limited amount of fine tuning.

Riutta et al (2007) measured GPP of mixed peatland vegetation in a nutrient-poor fen (peatland with a shallow water-table fed by lateral inflows). They isolated *Sphagnum* GPP but not $R_p$ or NPP, at a field plot during the





growing season. This was not a time series such as is common in ecosystem $CO_2$ eddy-covariance measurements, but rather as isolated functions of GPP against PAR, temperature and WTP. Our model was, therefore, run using fixed environmental model-inputs (PAR, temperature and WTP) to reproduce *Sphagnum* GPP functions against temperature and PAR, to simulate the Riutta data. One change was required to the LAI parameter to accommodate

the different *Sphagnum* species to the earlier calibration data, explained in Appendix B.2. We note that this curve-fitting factor may have corrected more than just the LAI differences between *Sphagnum*-species, which is further addressed in section 4. The model output is therefore compared to the Riutta test-data in figure 2.

Also, non-desiccation-stressed full daylight canopy dark respiration within the TRIFFID *Sphagnum* model was

between 0.4 µmol m$^{-2}$s$^{-1}$ (LAI=1, fen *Sphagnum*) and 0.7 µmol m$^{-2}$s$^{-1}$ (LAI=3, bog *Sphagnum*). $V_{max}$ is a fixed value in the model of 16 µmol m$^{-2}$s$^{-1}$ from equation A.11, being proportional to the leaf nitrogen content value, nl (kgNm$^{-2}$), which is a fixed value in the model. These values compare well with Williams & Flanagan's (1998) average daylight summer laboratory measurements of a truncated bog/poor fen *Sphagnum* sample with an uncertain LAI, giving a value for canopy dark respiration of 1.0±0.3 µmol m$^{-2}$s$^{-1}$ (at one standard deviation) and a value for $V_{max}$

of 14.0±4.0 µmol m$^{-2}$s$^{-1}$.

Thus, using available field or laboratory data, our *Sphagnum* PTF model visually reproduces the shape of laboratory-derived *Sphagnum*-$CO_2$ exchange curves, but with some clear residuals of up to about 40% against WTP and temperature, and up to +100% against PAR. The significant amount of detailed work performing the above

calibrations and testing is described in Appendices 1 & 2.



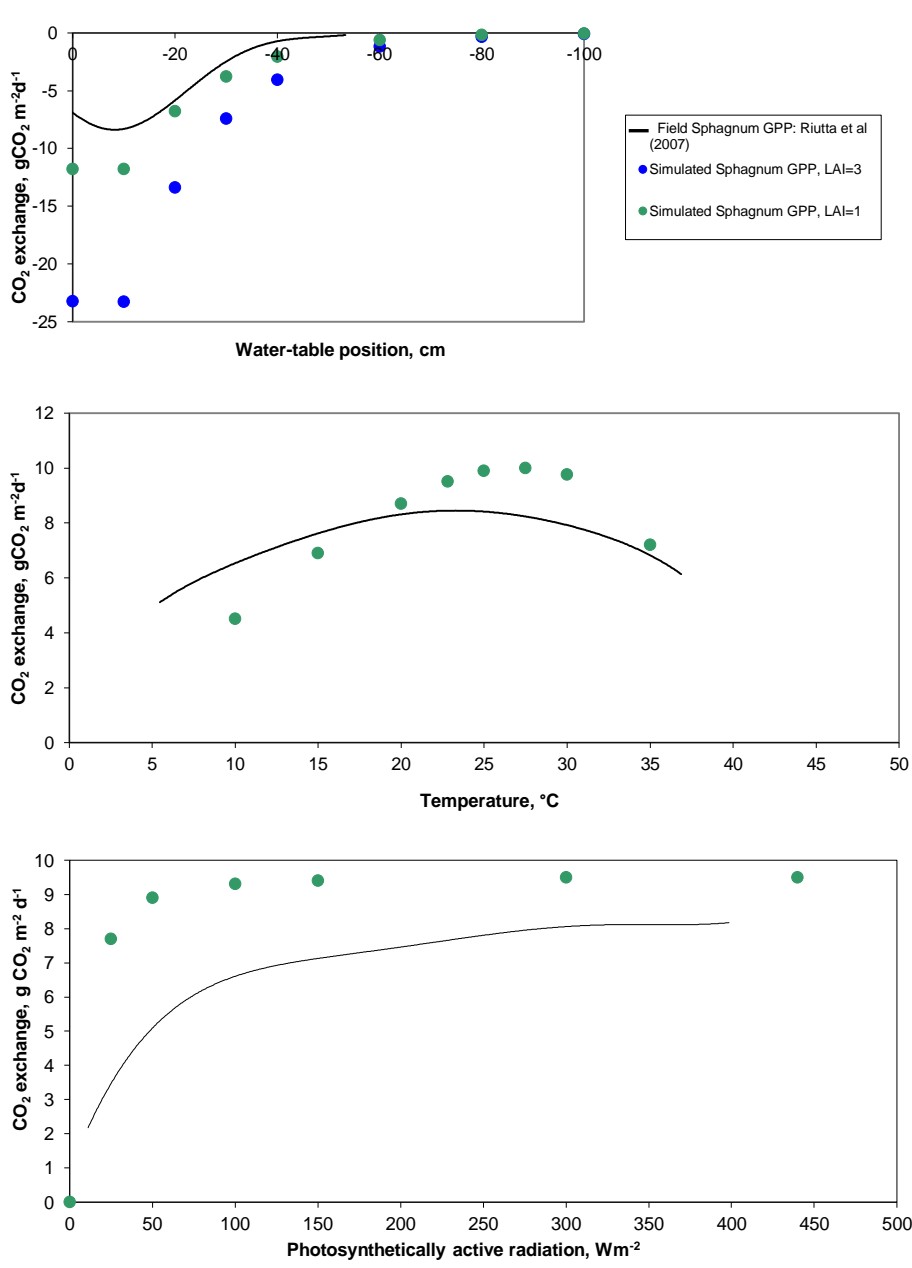

**Figure 2. Model outputs of GPP against WTP, temperature, and PAR, compared to field data from a *Sphagnum*-dominated peatland obtained by Riutta et al (2007).**





## 4. Discussion and Conclusions

A new process-based non-vascular-PFT model has been developed. *Sphagnum* mosses exhibit characteristic curves of $R_p$, GPP and NPP against WTP, PAR and temperature that are very markedly different to vascular plants (see Riutta et al. 2007). The new *Sphagnum* model reproduced the overall shape of these *Sphagnum* curves without any calibration. We believe this is the first time any process-based non-vascular plant photosynthesis model has reproduced these distinct curves. The model required only $r_g$ adjusting in order to calibrate it to the data from Strack et al (2009). Only a simple adjustment of the LAI parameter, being well supported by the literature to accommodate a different *Sphagnum* species, allowed our model to reproduce Riutta et al's (2007) GPP curves against WTP and temperature within about ±40% of the field-measurements. The fit to Riutta's PAR curve was less accurate at between +20% and +100% residual. We did not attempt any closer fitting here because this was intended to be test data to validate the initial calibration against the Strack data. There is insufficient test data to support statistical analysis of the residuals, with only a small number of features in the test-data curves. However, we also note that no previously published process-based models that we could find of *Sphagnum* photosynthesis (Druel et al. (2017), Qiu et al. (2018), Wania et al. (2009a, 2009b) and Yurova et al. (2007)) made any comparison to isolated field *Sphagnum*-NPP data, so our model is a first attempt at a greater level of process-detail together with a direct comparison to this field data. Of course, more field data would be welcome to improve confidence and to help refine the model.

Furthermore the use of LAI and $r_g$ as curve-fitting factors may accommodate a broad range of uncertainty elsewhere in the fitting of the model parameters, including, for example, varying photosynthetic efficiency of different *Sphagnum* species, internal near-surface water storage, transpiration, and varying $CO_2$ transport pathways available through pore-water and the atmosphere to the *Sphagnum* capitulum (Proctor et al, 1992; Lamers, 1999; Limpens et al 2008). These are little understood and therefore not candidate functions to be included in the *Sphagnum* PFT model in the foreseeable future.

Improved *Sphagnum* NPP models offer the possibility of simulating the accumulation of peatland plant litter with varying decomposition recalcitrance values and hydraulic properties (Wania, et al. 2009a), which influence the stability of the peat soil carbon store (Belyea 2009; Frolking et al. 2001). These two points are consistent with the observation of complex feedbacks between plant physiology and soil moisture (Bevan 2012) and Laurent et al.'s





(2004) view that, in order to improve the accuracy of existing vegetation carbon cycling simulations, simple climate-correlation (Brovkin et al. 1997; Prentice et al. 1992) may be enhanced by the inclusion of non-climate factors including plant-soil feedbacks. Additional parameters to be integrated within the model could be, for example, nutrient content (the nitrogen cycling is already represented in some other bryophyte models, e.g.

Euskirchen et al. 2009), intra-annual flood-frequency as already simulated explicitly in LPJ-WHY (Wania, et al. 2009b) and implicitly in JULES (O'Connor et al. 2010), and the influence of small-scale land topography on vegetation assemblage functions at higher scales (Baird et al. 2009; Belyea & Baird 2006; Sonnentag et al. 2008; Waddington & Roulet 1996).

Additionally, field and laboratory measurements show that *Sphagnum* $V_{max}$ has significant seasonal variation. All of the literature sources used to calibrate the photosynthetic and respiratory function of our model collected their data from *Sphagnum* samples taken during summer (see Table 5). Hence, the seasonal pattern of *Sphagnum* photosynthesis cannot be produced by this initial version of the TRIFFID *Sphagnum* model. Williams & Flanagan (1998) measured reduced $V_{max}$ values in spring and autumn, at half the summer value. This can be ascribed to a

summer maximum concentration of photosynthesising tissue (Skre & Oechel, 1981). Leaf nitrogen content is strongly correlated with plant photosynthesising capacity (Woodward 1994). Rice (2000) measured seasonal changes in the capitulum carbon to nitrogen ratio in three *Sphagnum* species that occupy contrasting niches on a temperate wetland in North Carolina. This showed a clear trend of increasing capitulum nitrogen concentration compared to carbon, which is strongly indicative of nl, leaf nitrogen concentration, between March and September,

and suggests that nl in *Sphagnum* should follow a seasonal pattern with a maximum value in summer. Therefore, the introduction of seasonally varying nl in place of the current fixed value in the model, is a candidate for future simulation of seasonal changes to *Sphagnum* $V_{max}$. Furthermore, while the quantum efficiency value in this model has been maintained at the C3 value of 0.08, used in JULES vascular PFTs, Kangas et al (2014) have measured quantum efficiency values for *Sphagnum* between 0.09 and 0.12 seasonally (with a minimum in July) and across

different *Sphagnum* species, but not correlated with local environment variables.

While the main focus of this paper has been the development of the new TRIFFID *Sphagnum* PFT model, the model we have produced is also readily suited to simulate true mosses, which play an important role in boreal ecosystems overlying unsaturated soils (Bond-Lamberty et al. 2004; Wieder, et al. 2006, Beringer et al 2001; Lindo




& Gonzalez 2010; Street et al. 2013). However, true mosses generally hydrate due to capillarity in their lower portions, distillation from a nocturnal temperature inversion of the underlying soil surface in their upper portions, or direct interception of precipitation (Carleton & Dunham, 2003; Lindo & Gonzalez 2010). This would require the replacement of the empirical water-uptake equation (point 1 in table 2) with a function reflecting this different

moisture-uptake mechanism, which is a relatively minor modification to the new *Sphagnum* non-vascular PFT. Precise quantification of this function does not yet appear to have been made empirically or theoretically, and this would be an important next step in enabling the adaptation of the current model to true mosses.

The most important outcome of this study is to have, for the first time, computationally solved the two-way

simultaneous function that governs photosynthesis in the non-vascular *Sphagnum* plant, thus simulating photosynthetic and respiratory functions of *Sphagnum*, which differ significantly from equivalent vascular vegetation functions, reproducing visually the NPP functions of *Sphagnum* samples. In other words we have developed, for the first time, a *Sphagnum* NPP model that resembles the internal function of the actual *Sphagnum* NPP mechanism, at a higher level of detail than existing PFT models such as those produced by Druel et al.

(2017), Qiu et al. (2018), Wania et al. (2009a, 2009b) and Yurova et al. (2007). Our work may therefore also contribute to a general photosynthesis model for *Sphagnum* beyond the immediate needs of DGVMs. It has been implemented in an existing latest-generation DGVM at a consistent level of detail and computational efficiency with the DGVM, but not coupled with the land-surface-functions of the DVGM such as hydrology and soil development, so it has not been possible to compare outputs to ecosystem eddy-covariance time-series.

Nevertheless even as it stands, the new model can be a tool to examine the carbon cycling patterns of *Sphagnum* under different climates.





## Appendix A. Simulation of net leaf photosynthesis

### A.1    Adaption of TRIFFID's C3 Photosynthesis Function to *Sphagnum*

*Sphagnum* exhibits the C3 carbon fixation photosynthesis pathway (Loisel, et al. 2009; Price, et al. 1997; Proctor et al. 1992). The TRIFFID vascular PFT with C3 carbon fixation has been adapted to replicate the non-vascular *Sphagnum* photosynthesis function. This involved replacing the stomatal regulation of gas exchange in the model with a fixed leaf conductance value, and removing the ability to extract water from deep in the soil using a vascular root system.

The TRIFFID functions that were adapted to newly simulate *Sphagnum* physiology are summarised here from Best et al. (2011) and Clark et al. (2011) (equation numbers with asterix). In TRIFFID, net leaf photosynthesis A (mol $CO_2$ m$^{-2}$s$^{-1}$) is computed using the solution to equations A.1 to A.6. GPP is the gross leaf photosynthesis, $\beta$ is the desiccation stress factor, a unit-less value between 0 and 1, simulated for vascular plants using equation A.6, where $\Theta$ is the mean soil water concentration in the root zone, and $\Theta_c$, $\Theta_w$ are the PFT-specific critical and wilting soil moisture concentrations respectively. Cox et al. (1998) showed that the simple desiccation-stress function in equation A.1 improved the fit of a vascular vegetation model for a minimal number of extra parameters. $R_d$ is dark respiration (see section 1), which is a linear function of $V_{max}$ and $f_T$. $c_{atm}$ is the atmospheric $CO_2$ partial pressure (Pa), and $c_i$ are the rubisco- and light-limited potential internal leaf $CO_2$ partial pressures. $O_a$ is the atmospheric oxygen ($O_2$) partial pressure (Pa). R is the ideal gas constant, T* is the leaf surface temperature (K), $\Gamma$ is the $CO_2$ photorespiration compensation point (Pa) where photosynthesis balances respiration. In TRIFFID this is simulated as a function of $O_a$ and the Rubisco specificity of $CO_2$ relative to $O_2$, $f_T$ $f_T$ , equations A.7 and A.8. $\omega$ $\omega$ is the leaf scattering coefficient for PAR, for which TRIFFID assumes $\omega$ $\omega$ = 0.15 for C3 plants, and the value 0.08 in equation A.4 represents the quantum efficiency of C3 plants, $\alpha$. Equation A.2 is the $CO_2$ diffusion equation.

For vascular plants exhibiting C3 photosynthesis, TRIFFID solves equations A.2 to A.5 as three simultaneous functions of leaf surface gas conductance (equation A.2), gross photosynthetic rate, GPP, and stomatal closure, which is a function of the leaf humidity deficit – see Table 2 (Cox et al 1998). GPP is calculated using a smoothed minimum of GPP$_r$ , GPP$_l$ , and GPP$_e$ (equation 1), which are GPP under photosynthetic-enzyme (Rubisco) limitation, light limitation, and photosynthetic-electron-transport limitation respectively, equations A.3 to A.5,




where Clark et al. (2011) in TRIFFID follows Sellers et al. (1996) and Cox et al. (1999). The three unknowns that are resolved, by the existing TRIFFID vascular PFT function, are A, $c_i$, and $g_s$.

$$A = (GPP - R_d)\beta \tag{A.1*}$$

$$A = \frac{g_s}{1.6RT_*}(c_{atm} - c_i) \tag{A.2*}$$

$$GPP_r = V_m \left\{ \frac{c_i - \Gamma}{c_i + K_c(1 + 0_a/K_0)} \right\} \tag{A.3*}$$

$$GPP_l = 0.08(1 - \omega)I_{par} \left\{ \frac{c_i - \Gamma}{c_i + 2\Gamma} \right\} \tag{A.4*}$$

$$GPP_e = 0.5V_m \tag{A.5*}$$

$$\beta_{vascular} = \left\{ \begin{array}{ll} 1, & \theta > \theta_c \\ \frac{\theta - \theta_w}{\theta_c - \theta_w}, & \theta_w < \theta \leq \theta_c \\ 0, & \theta > \theta_w \end{array} \right\} \tag{A.6*}$$

$$\Gamma = \frac{O_a}{2\tau} \tag{A.7*}$$

$$\tau = 2600 f_T(0.57) \tag{A.8*}$$

$$f_T(q_{10}) = q_{10}^{0.1(T_c - 25)} \tag{A.9*}$$

$V_m$ is calculated in equation A.10, and is a function of the lower and upper PFT–temperature production limits, $T_{upp}$ and $T_{low}$ and Q10 temperature dependence, $f_T$, the non-temperature-dependent rate, $V_{max}$, that simulates nitrogen-limitation of C3 photosynthesis (Clark et al. 2011) and leaf surface temperature, $T_c$. $V_{max}$, equation A.11 is kept the same for *Sphagnum* and other C3 plants following the example of Yurova, et al. (2007), and is a linear function of leaf nitrogen concentration, nl (kgNm$^{-2}$).

$$v_m = \frac{v_{max} \ f_T(2.0)}{\left(1 + e^{0.3(T_c - T_{upp})}\right)\left(1 + e^{0.3(T_{low} - T_c)}\right)} \tag{A.10*}$$

$$v_{max} = 0.0008nl \quad \text{for C3 plants including } Sphagnum \tag{A.11*}$$



## A.2 Gas Exchange Formulation for *Sphagnum* PFT

$g_{s\,Sphagnum}$ applied to equation A.2 was set to 1.6*0.0237*0.07 ms$^{-1}$ and is unregulated, reflecting constant exposure of the *Sphagnum* photosynthesising tissues to the atmosphere. The 1.6 is unitless and accounts for differing molecular diffusivity of $CO_2$ and water vapour in air, and 0.07 mol m$^{-2}$s$^{-1}$ is the maximum $CO_2$ conductance measured in *Sphagnum* by Williams & Flanagan (1998). The value of 0.0237 m$^3$mol$^{-1}$ is a fixed non-temperature-dependent amount for the molar volume. It is not clear that Williams & Flanagan's (1998) $CO_2$ conductance in mol m$^{-2}$s$^{-1}$ within the living *Sphagnum* tissue would be constant with respect to temperature. Therefore, for simplicity, the molar conversion avoids temperature-dependence. This is in order to avoid adding complexity (adding temperature dependence to the *Sphagnum* volumetric gas absorption function) which may make the analysis of this initial model difficult. Williams & Flanagan (1998) ascribed lower values to desiccation-stress, which were not applied in our model because desiccation-stress is simulated independently using the desiccation stress factor, β, defined for our *Sphagnum* model in Table 2. This is based on field observations from Strack et al. (2009) (capitulum water content against WTP), Murray et al. (1989) and Riutta et al. (2007) (maximum *Sphagnum* photosynthetic production value occurs when the water table is just below the ground surface). It is also similar to the approach of Wania et al. (2009b), Yurova et al (2007) for their *Sphagnum* models. Only two simultaneous functions needed solving to simulate *Sphagnum* photosynthesis, represented by equations A.2 to A.5, leaving out completely the stomatal conductance term referred to in Table 2. Therefore, we changed the three-way simultaneous function described in appendix A.1 for vascular plants, to a two-way simultaneous function for non-vascular *Sphagnum*. The two unknown variables are A, and $c_i$. Combining equations A.1 and A.2 with equations A.3 (rubisco-limited net leaf photosynthesis), and then combining equations A.1 and A.2 with equation A.4 (light-limited net leaf photosynthesis), given that $g_s$ is now fixed, gives equations A.12 and A.13:

$$0 = \frac{\alpha}{V_m} c_{ri}^2 + \left(\frac{\alpha}{V_m}(K - c_{atm}) - R_d + 1\right)c_{ri} - \left(\frac{\alpha}{V_m}Kc_{atm} + KR_d + \Gamma\right) \tag{A.12}$$

$$0 = \frac{\alpha}{\gamma} c_{li}^2 + \left(\frac{\alpha}{\gamma}(2\Gamma - c_{atm}) - R_d + 1\right)c_{li} - \left(\frac{\alpha}{V_m}Kc_{atm} + KR_d + \Gamma\right) \tag{A.13}$$

where $\alpha = \frac{g_s}{1.6RT^*\beta}$, $\gamma = 0.08(1 - \omega)I_{par}$, $K = K_c\left(1 + \frac{0_a}{K_0}\right)$. α, γ and K are terms to simplify equations A.17 and A.18.



In the new *Sphagnum* PFT, leaf-surface conductance is fixed and the rate of photosynthesis varies. Therefore, under different limited rates of photosynthesis, we have split $c_i$ (internal leaf $CO_2$ partial pressure) to distinguish it between $c_{ri}$ (under rubisco-limitation) and $c_{li}$ (under light-limitation). In the equations for the prior vascular TRIFFID PFT, $c_i$ was instead the same value under both limitations. (This does not affect the use of equations 1 and 2 to calculate $g_s$ in the new *Sphagnum* PFT since these equations are not used to calculate $g_s$, which has a fixed value instead). The quadratic solutions to equations A.12 and A.13 give a positive and a negative value for $c_{ri}$ and $c_{li}$. The positive solutions for $c_{ri}$ and $c_{li}$ were substituted respectively into equations A.3 and A.4 to give unique values of $GPP_r$ and $GPP_l$. The question of whether to use the positive or negative quadratic solution was answered in the following way. The standard quadratic solution is:

$$c_i = \frac{-B \pm \sqrt{B^2 - 4AD}}{2A} \qquad \text{A.14}$$

Applying equation A.14 to equations A.12 and A.13, where all the additional variables are positive numbers:

$$A_r = \left\{ \frac{\alpha}{V_m} \right\}, A_l = \left\{ \frac{\alpha}{\gamma} \right\} \qquad \text{A.15}$$

It follows that $A > 0$ in both cases.

$$B_r = \frac{\alpha}{V_m}(K - c_{atm}) - R_d + 1, \qquad B_l = \frac{\alpha}{\gamma}(2\Gamma - c_{atm}) - R_d + 1 \qquad \text{A.16}$$

It follows that $B > 0$ or $B < 0$ are both possible.

$$D_r = -\left\{ \frac{\alpha}{V_m} K c_{atm} + K R_d + \Gamma \right\}, \qquad D_l = -\left\{ \frac{\alpha}{\gamma} 2\Gamma c_{atm} + 2\Gamma R_d + \Gamma \right\} \qquad \text{A.17}$$

It follows that $D < 0$, therefore $-4AD > 0$, and $\sqrt{B^2 - 4AD} > B$. So in order for the quadratic equation to give positive values for $c_{ri}$ and $c_{li}$, then its solution must take the form of equation A.18.

$$c_{ri}, c_{li} = \frac{-B + \sqrt{B^2 - 4AD}}{2A} \qquad \text{A.18}$$

The smoothed minimum value for GPP is then calculated using the existing smoothing algorithm in TRIFFID (Clark et al. 2011) from the unique solutions to equations A.3 through A.5. This creates a new version of the photosynthesis process in Table 2 (Foley et al. 1996, Clark et al. 2011) adapted to accommodate the differences in the *Sphagnum* organism described in section 2.



## Appendix B. Use of Shaping Parameters

### B.1 Simulation of *Sphagnum* plant respiration

We compared the equations that simulate respiration in TRIFFID with the equivalent LPJ equations from Yurova et al. (2007), in order to correctly apply the *Sphagnum* field-derived parameterisations from LPJ's respiration equations to TRIFFID.

In TRIFFID, $R_p$ is a function of $R_d$, dark leaf respiration, in equations B.1 and B.7, which is simulated explicitly. Dark respiration is the plant respiration that occurs in cell mitochondria as opposed to photosynthesising cell components. The former provides energy for plant function whereas the latter reduces specifically photosynthetic efficiency (Allaby, 2006).

Equations B.1 and B.6 have the same form, simulating $R_p$, in TRIFFID ($R_{pT}$ - units of $kgCm^{-2}s^{-1}$) and LPJ-GUESS ($R_{pLPJ}$ - units of $kgCm^{-2}d^{-1}$) respectively.

TRIFFID

$$R_{pT} = r_{gT}(\Pi_{GT} - R_{pmT}) + R_{pmT} \qquad \text{B.1}$$

$$\Pi_{GT} = 0.012(A_C + R_{dc}\beta) \qquad \text{B.2}$$

$$R_d = f_{dT} \cdot V_{max} f_T(2.0) \qquad \text{B.3}$$

$$R_{dc} = \sum(R_d) \qquad \text{B.4}$$

$$A_c = \sum(A) \qquad \text{B.5}$$

LPJ-GUESS

$$R_{pLPJ} = r_{gLPJ}(\Pi_{GLPJ} - R_{pnon-photLPJ} - R_{pphotLPJ}) + (R_{pnon-photLPJ} + R_{pphotLPJ}) \qquad \text{B.6}$$

$R_{pmT}$ is plant maintenance respiration in TRIFFID; this is a function of canopy dark leaf respiration from equation B.7 . The first right-hand term in equations B.1 and B.6 simulate growth respiration that is proportional to canopy NPP, ($\prod_G$ -$R_{pm}$) in TRIFFID, and ($\prod_G$ -$R_{pnon-photLPJ}$-$R_{pphotLPJ}$) in LPJ. $r_g$ is the growth respiration coefficient, and





$\prod_G$ is PFT-canopy GPP rate (kgC m$^{-2}$s$^{-1}$). The second right-hand terms simulate maintenance respiration. $R_{pnon\text{-}phot}$ and $R_{pphot}$ in LPJ-GUESS refer to respiration in photosynthetic and non-photosynthetic tissues. $R_{dc}$ and $A_c$, in TRIFFID, are canopy dark respiration and canopy net photosynthesis, upscaled from leaf dark respiration, $R_d$, and net leaf photosynthesis, A, using the 'big-leaf' method (see Clark et al. 2010; Clark et al. 2011; Cox, et al. 1999), in simplified equations B.4 and B.5. We used this method for simplicity and execution-speed; TRIFFID also has more sophisticated and less computationally efficient leaf-area upscaling methods that were not used. $R_{pm}$ in TRIFFID is an equivalent term to ($R_{pnon\text{-}phot} + R_{pphot}$) in LPJ-GUESS.

TRIFFID

$$R_{pmT} = 0.012 \cdot R_{dc} \left\{ \beta + \frac{N_r + N_s}{N_l} \right\} \qquad \textbf{B.7}$$

$N_l$, is the leaf nitrogen content. $N_s$ and $N_r$ are the stem and root nitrogen contents. We split Equation B.7 to compare $R_{pphotT}$ against $R_{pphotLPJ}$.

TRIFFID

$$R_{pmT} = R_{pnon-photT} + R_{pphotT} \qquad \textbf{B.8}$$

$$R_{pphotT} = 0.0012 \cdot R_{dc} \cdot \beta \qquad \textbf{B.9}$$

LPJ-GUESS

$$R_{pphotLPJ} = f_{dLPJ} \cdot V_{max} \qquad \textbf{B.10}$$

Whereas in LPJ $R_{pphotLPJ}$ is non-temperature dependant (equation B.10), in TRIFFID $R_{pphotT}$ has a Q10 temperature-dependence function (equations B.3, B.4, B.8 and B.9), but $f_d$, the unitless dark respiration scaling factor, in both LPJ and TRIFFID is applied simply to $V_{max}$ (equations B.3 (TRIFFID) and B.10 (LPJ), so the value $f_{dT} = f_{dLPJ} = 0.03$ for *Sphagnum* has been adopted for TRIFFID unchanged from LPJ.





Respiration within non-photosynthetic tissue again shows a similar form for LPJ-GUESS and TRIFFID.

TRIFFID

$$R_{pnon-photT} = 0.012 \cdot R_{dc}\left\{\frac{N_r+N_s}{N_l}\right\} \tag{B.11}$$

LPJ-GUESS

$$R_{pnon-photLPJ} = r_m \cdot k_r \cdot g(T_{air})\left\{\frac{M}{cn}\right\} \tag{B.12}$$

In equation B.12 the combination of the shaping parameter $r_m$, $k_r$ and $g(T_{air})$ represent a similar term to $R_{dc}$ in TRIFFID. $k_r$ (day$^{-1}$) is the respiration rate for a 10ºC baseline, and $g(T_{air})$ is a temperature response function (Yurova, et al. 2007). M is total biomass (kgCm$^{-2}$), cn is the C:N ratio. The term $\frac{M}{cn}$ represents the N content of the plant against the unitless mass-ratio term in TRIFFID of $\left\{\frac{N_r+N_s}{N_l}\right\}$. The shaping parameterisation for $R_{pnon-photT}$ in

TRIFFID comes again from the dark-respiration formulation in equation B.3, omitting the $r_m$ shaping parameter from LPJ. The differing mass-ratios in these equations have not been reconciled between TRIFFID and LPJ-GUESS. Therefore the shaping parameter $r_{gT}$ was calibrated in place of the two parameters $r_{gLPJ}$ and $r_{mLPJ}$ in LPJ-GUESS in order to fit the output curves in Figures 1 and 2. It is notable that $r_{gT}$ with a value of 0.1 is lower than the similar $r_{gLPJ}$ (0.5) or the value of $r_{gT}$ previously used in TRIFFID, 0.25 (Clark et al. 2011) for vascular plants,

and this implies that the values of $f_{dT}$ and $r_{gT}$ could be more tightly constrained, when more calibration data is available.

In summary, the value of $f_d$ for TRIFFID in Table 4 has been established for *Sphagnum*, which is the same as the equivalent parameter in LPJ-GUESS. In this version of the TRIFFID *Sphagnum* model, parameter $r_g$ is used as a

shaping parameter.

### B.2 *Sphagnum* Leaf Area Index

The model outputs did not fit the field-data of GPP against WTP from both Riutta et al (2007) and Strack et al (2009), without the LAI parameter being changed.  Bond-Lamberty & Gower (2007) measured moss-LAI in a

forested peatland, and found that *Sphagnum* and other boreal mosses have LAIs between 1 and 10.  Glenn et al. (2006) measured the LAI of *Sphagnum* in a nutrient-poor fen as 1. Rydin et al (2006) noted that *Sphagnum* species



that form lawns have lower shoot-density than those that form hummocks. The former include the species examined in Riutta et al (2007), the latter include the species in Strack et al (2009). Therefore LAI=1 (equivalent to lower shoot density) was applied to fit the model to data from Riutta et al (2007) (lawn/fen species with lower shoot-density) and this parameter was changed to LAI=3 (equivalent to higher shoot-density) in order to fit the data from

Strack et al (2009) (hummock/bog species with higher shoot-density). In addition, it is also plausible that a different desiccation stress function is required for these different *Sphagnum* species, with different parameters for the equation that defines $\beta_{Sphagnum}$ in Table 2, but different stress functions have not been developed here.

**Code Availability**

We have modified subroutines in the Surface section of the JULES v2.1 model code. This changed code is available
at the DOI link in Coppell (2019), for which we obtained permission from the Met Office. We must emphasise that JULES v2.1 is not the latest version of JULES and differs in much of its functionality from the latest version. Permission and license-information to use the whole JULES model is available from http://jules-lsm.github.io/access_req/JULES_access.html , from where the complete original JULES v2.1 model may be requested by email.

**Author Contribution**

Richard Coppell – developed the simultaneous-function part of the model with EG and designed and implemented all other parts of the model, tuned and tested, and co-wrote the paper with JH.
Emanuel Gloor – developed the simultaneous-function element of the model with RC and proof-read the paper.
Joseph Holden – supervised and co-wrote the paper with RC.

**Competing Interests**

The authors declare that they have no conflict of interest.



**Acknowledgements**

We would like to thank Andy Baird, water@leeds, School of Geography, University of Leeds, UK, for influential advice on the early direction of the research and the literature base. The following were of great assistance in using the JULES model: Stephen Sitch, water@leeds, School of Geography, University of Leeds, UK (currently at
5 Department of Geography, University of Exeter, UK); Chris Jones, The Hadley Centre for Climate Prediction and Research, Exeter, UK; Doug Clark, The Centre for Ecology and Hydrology, Wallingford, UK; Alan Real & Dureid El-Moghraby, Information Systems Services, University of Leeds, UK. We are also grateful to Martin Best, The Hadley Centre for Climate Prediction and Research, for giving limited permission to make available our changed JULES code on a DOI link for this paper. Additional thanks go to Sergey Venevsky, School of Geography,
University of Leeds, UK (currently at Center for Earth System Science, Tsinghua University, China) and Katherine Arrell for additional supervision. Finally, thanks go to Alice Noble, University of Leeds; Lorna Street, University of Edinburgh and Rita Wania, who was previously at the University of Bristol, for discussions regarding *Sphagnum* and moss carbon cycling. R. Coppell was funded by a NERC scholarship NE/F008341/1 supported by the Met Office Hadley Centre as CASE partner, which led to the technical support received from Chris Jones and Doug
Clark, above. Richard Coppell also acknowledges his cousin, David Rogers, for proof-reading and the Additional Learning Support department at Leeds City College for employment whilst continuing this work, and in particular Steve Brown. This work was undertaken on ARC1, part of the High Performance Computing facilities at the University of Leeds, UK.





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
