# Peer review of "A process-based *Sphagnum* plant-functional-type model for implementation in the TRIFFID Dynamic Global Vegetation Model"

_Geoscientific Model Development, 2019_

## Short Comment (SC1) · 26 Jul 2019

Dear authors,

in my role as Executive editor of GMD, I would like to bring to your attention our Editorial version 1.2:

https://www.geosci-model-dev.net/12/2215/2019/

This highlights some requirements of papers published in GMD, which is also available on the GMD website in the 'Manuscript Types' section:

http://www.geoscientific-model-development.net/submission/manuscript_types.html

In particular, please note that for your paper, the following requirement has not been met in the Discussions paper:

- "The main paper must give the model name and version number (or other unique identifier) in the title."

You neither provide a version number nor a name/acronym for the published model (the process-based Sphagnum plant-functional-type model), nor do you provide the information in the title that this is all implemented in JULES. Therefore I suggest to change the title to somethiing like:

"A process-based Sphagnum plant-functional-type model (PBSPFT v0.0) for implementation in the TRIFFID Dynamic Global Vegetation Model in Jules v2.1"

At least the information on Jules v2.1 must be added.

Finally note, that according to our new Editorial (v1.2) all data and analysis / plotting scripts should be made available.

Yours,

Astrid Kerkweg

---

## Short Comment (SC2) · 1 Aug 2019

Thanks for this. Would I need to make the modifications and then submit a new version of the paper for this ongoing discussion?

Is it okay if I do the following:

(1) Amend the title as per your suggestion. (2) Put all data that is used in the figures in a table in an additional appendix section, and explain how it was plotted, and also explain in detail how it was extracted from the model. (in short - the variables were extracted from the JULES model using extra code-lines in the model, into a csv file,

which was then imported and plotted in Excel, so if I explain this more precisely). (3) in the same appendix section, I will put all the data that I extracted from other papers for comparison, and explain how I extracted it (I used 'datathief' software to read the data from their graphs, just to get their curves).

Thanks again,

Richard Coppell

―――――――――――――――――――

---

## Referee Comment (RC1) · Anonymous Referee #1 · 18 Dec 2019

This study by Copper et al. presents a new Sphagnum plant functional type model designed for implementation in the TRIFFID dynamic vegetation model. As far as I can judge, such a study is novel and is worth being published. The main novelty is the extension of photosynthesis/growth model to non vascular plants. The text is relatively clear and well written. However, there are a certain number of points that must be considered before publication, including some mistakes in the equations that must be corrected.

Major points:

(1) The $\beta$ factor in Table 2 and equation A.1 multiplies the assimilation rates. This

assumes that photosynthesis responds directly to soil water stress. Personally I think that, at least for vascular plants, this response should be indirect, i.e., it is a response to the progressive closure of stomata, which themselves react to decreasing soil water. So, for vascular plants, it is the stomatal conductance gs that should be affected by the $\beta$ factor. The authors refer to Cox et al. (1998) to justify this multiplication of the assimilation rate by $\beta$. Cox et al (1998) indeed used a $\beta$ factor (very similar, although slightly different mathematically), but this factor was applied to the stomatal conductance (Jarvis model), not to the assimilation rate. The authors should justify their procedure. It implies that when soil water decreases to the wilting point $\theta$w, the $\alpha$ parameter grows to infinity, whereas it would tends towards zero in the case the $\beta$ factor multiplies gs. So, the mathematical solution may be impacted. For non vascular plants, gs may be considered as independent of soil water (as the authors assume), although Williams and Flanagan (1998) reported a dependence on water table depth (as mentioned by the authors in appendix 2). Also, what is the critical soil water $\theta$c? Does it correspond to the field capacity, as in Cox et al. (1998)? It is reasonable to assume that stomatal conductance is progressively lowered (at least for vascular plants) when soil water decreases below field capacity. But it is not really reasonable to assume that plant desiccation will start at field capacity. For vascular plants, this normally starts at $\theta$w. So, why multiply assimilation by the $\beta$ factor? For non-vascular plants, it may be different, desiccation may start earlier, but a justification of the procedure used here is needed and it should be based on an explanation of the processes (or at least on process-based hypotheses).

(2) There are mistakes in equations A.12, A.13 and A.16 and A.17. These equations are not dimensionally correct. In these equations, Rd should be divided by Vm (eq. A.12 and A.16) or by $\gamma$ (eq. A.13 and A.17). Moreover, the last term of eq. A.13 is also wrong: it cannot contain K that comes from eq. A.1, since A.13 does not correspond to the Rubisco limitation case. Please, correct these equations. If these equations are also incorrect in the code, then the code should be modified and rerun. But possibly it is just a problem of the manuscript. Please check.

[Figure]

(3) Your validation in Figure 1 is not convincing. With respect to Strack et al. (2009) you have modified the respiration curve. Indeed Strack et al. report ecosystem respiration (ER) and net ecosystem exchange (NEE), but you need Sphagnum autotrophic respiration (Rp) and net primary productivity (NPP), because your model does not contain soil respiration (i.e., peat decomposition) and vascular plant respiration. For that reason, you are making a correction to the respiration curve of Strack et al. (2009), that you take as constant, as explained on page 18. This explanation is not convincing. For instance, you just discuss vascular plant respiration, but not heterotrophic respiration, which may vary strongly with water table depth. I would recommend removing the Rp and NPP curves in figure 1, and thus validating only GPP from these data.

Minor points:

- p. 12, definition of gs: it should be specified that this conductance is for water vapour - p. 13, definitions of Nl, Nr and Ns: units should be mol N m-2, not mol CO2 m-2 s-1 - p. 15, table 3: description and references should appear in different columns for clarity. Also description may be slightly expanded (some of the parameters are very specific to TRIFFID) - p. 18, lines 16-19: you should provide somewhere in the paper the climatic inputs you used in the model, since these are fixed. We need to know more about your model inputs. - Figure 1, p 19: use more contrasting colours than dark blue and black - Figure 2, p 21: the difference between model and data is sometimes quite important. May need more comments. - P. 25, line 13: notation capital $\Theta$ should be replaced by lower case $\theta$ - p. 25, line 20: fT and $\omega$ both occur twice - both Vmax and Vm are used for Vc,max. Is there a difference? Should be harmonized. - p. 30 line 18: "dependent" instead of "dependant" - p. 31 line 27: "…that different desiccation stress functions are required…" instead of "…that a different desiccation stress function is required…"

Please also note the supplement to this comment:
https://www.geosci-model-dev-discuss.net/gmd-2019-51/gmd-2019-51-RC1-
supplement.pdf

---

## Referee Comment (RC2) · Anonymous Referee #2 · 15 Oct 2020

This paper introduces into the TRIFFID model, a generically long-held ambition of many land surface and dynamic vegetation modeling groups - to treat the physiology of non-vascular plants (specifically spagnum moss), without stomatal control, in the same framework as that which is typically deployed to simulate the gas exchange of vascular plants. In that sense, this paper is a welcome addition to the literature and could, I imagine, form the basis of many developments that are based on top of these fundamental updates.

Data on the physiology of spagnum moss are difficult to come by, and an initial assessment of the validity of the model was made against available data.

In general, I found the discussion of the data sources quite difficult to follow through the paper, and also, the results section is missing a way of comparing the spagnum model to the standard photosynthesis scheme used in JULES-TRIFFID.

Further, I feel like the paper is not sufficiently 'polished' in tems of the clarity of the writing, nor the description of the model modifications, to merit publication at this stage. Most critically, the main text is missing a discussion of why and how the primary modifications to the photosynthesis scheme were made. It seems like the paper needs a few more iterations between coauthors before it reads clearly enough to those unfamiliar with the work or the surrounding literature.

Lastly, while the first author has gone to appropriate lengths to make the code for these modifications available, they are largely not useful without the full code infrastructure of the JULES model to place them in context. I appreciate very sincerely that this is not the fault of the authors, but it is in my view untenable to retai this state of affairs in 2020.

Specific Comments

P1 L10: I think it's not necessary to say 'more recent' DGVM, most models I know of have had sub daily physiology for at least 20 years.

P1 L13: The paper title says 'for implementation in' whereas this suggests it is already implemented in TRIFFID. Which is it?

P2 L1: in this paragraph the focus is on DGVMs, but these are a subset of land surface models that all could benefit from improved representation of mosses. i suggest making this less focused on DGVMs (given this is really a paper about representing physiology and not vegetation dynamics) and more on LSMs in general.

P2 L16: This section needs references expanding outside of the JULES/TRIFFID literature. P2 L31: A clear statement on the magnitude of how much carbon is derived from sphagnum would be useful hear. Earlier, you refer the half the carbon in Northern

peatlands, but an estimate of their spatial extent and/or carbon store would help frame the importance of the representation of sphagnum better. P2 L31- P3 L5: This section is a little muddled and could do with reorganization.

P3 L10: This paragraph states that this is in fact an existing moss physiology module in JULES, and then later that there is not. Really, a more focused discussion is needed on the deficiencies of the existing sphagnum representation.

P4 L1: Do you mean NEP here (given previous references to soil? ) P4 L8: The word 'assemblage' as used in this context is something I, for one, am not extremely familiar with. Can you define/explain it before using it here?

P4 L20: I do not think that the term 'PFT model' has any specific meaning here. P6 L23-29: This section on photosynthesis doesn't really tell us anything specific about moss, other than that it is a C3, which doesn't seem to need a whole paragraph?

P7 L2: Across what range does it vary between species?

P7 L6: And also dew?

P7 L7: Is there a point to add concerning the thermal insulation properties of the moss layer(s)? P7 L14: What -does- the model contain? That seems like an important thing to add before stating what it doesn't? Why not modify the canopy water storage terms to account for the moss properties? P7 L15: What is microform position?

P7 L21: Again, 'PFT model' is redundant here.

P7 L27: Indeed, but Prof. Cox is also the last author on the recent implementation of plant hydraulics in JULES (Eller et al. 2020) following from numerous other implementations of hydrodynamic schemes into other land surface models. I suggest you reframe the syncing with an explicit water uptake scheme as something to be incorporated into later versions, rather than digging any deeper into the defense of the previous empirical scheme.

P7 L28: Where was the sphagnum in question located? I feel like this needs a little more detail as it is the primary dataset employed here.

P8 L1: A fixed maximum value of what?

P8 L6: I'd say "for vascular plant in the TRIFFID model"

P8 L7: Did Williams and Flanagan use JULES/TRIFFID? Or is this a different approach altogether (in place of is a confusing word choice here).

P8 L9: Successful approaches to what? I'd be slightly more circumspect about attributing 'success' in the context of model elements of land surface models in the absence of very comprehensive and specific benchmarking of particular elements. Models have so many degrees of freedom that one can trivially get the right answer for the wrong reason a lot of the time.

P8 L25: A note on the potential pitfalls of used smoothed photosynthesis can be found in this recently published analysis by Walker et al. (2020)

P8 L20-27: This part is difficult to follow given that one has to skip between the table and the text. I'd recommend integrating the explanation of what you have done into the text and not using the table, which is a nce idea in principle but actually quite hard to follow, in particular wrt the mathematics of the solution of the photosynthesis/gs scheme. Why was it necessary to change the solution? I don't really understand from what is written here why simply adding a constant stomatal conductance does not suffice. Further, the references to Druel et al. and Dimitrov et al. need expansion here, given that this section really explains the key developments tht you have made.

P9 L3: How is LAI a model parameter? Do you mean LAImax? LAI is an output of TRIFFID, unless I'm very much mistaken? Also this paragraph could do with having the parameter table closer to it in the text, otherwise these names come rather out of the blue.

Table 2: Under the first heading "carbon gain and mass balance" is the description

of the Beta factor controlling stomatal responses to soil drying. This needs correcting and/or describing better. Table 2: "Photosynthesis Machine" should probably just be "Photosynthesis" Table 2; Why is respiration in this table when the modification of the parameters would more naturally exist in table 3?

Table 5: Why organize the information in this way? It requires one to shuttle backand forth between at least three sections at once to figure out what information is used where. I would reorganizt this in to section in the text by process, and perhaps keep table 3, (merged with 4?) but lose 2 and 5.

P 17 L 1: What criteria does a dataset need to meet to be useful for calibration and /or validation of this model?

P17 L8-10: I'm not sure what the 'correction' is that you are trying to describe here. P17 L 12: Is this 'correction' still trying to account for 20% non-sphagnum vegetation in the first dataset? This is pretty hard to follow in the absence of familiarity with these references (and given that representation of moss in DGVMs and LSM is essenttially a new topic, I think you can safely assume the majority of your readers will not be familiar with this literature).

P17 L20: This statement on NPP seems a little redundant.

Figure 1: It is generally customary to have model output as lines and field observations as points in these types of figures.

P18 L5: Reiterate what type of data are available fro Riutta et al.

P18 L6: It should not be especially hard to generate an RMSE, or R2 value? I appreciate that getting the general patterns of behaviour is a higher order concern, but it sees like it should be easier to just calculate some statistics than to argue why it is unnecessary to do so.

P18 L9: This section would be useful earlier on.

P19 L5: Which earlier data? This is making things unnecessarily hard on the reader. Also, LAI is not a parameter here, it is presumably an input, but I remain confused about why it is not an output of the fully TRIFFID carbon cycle physiology. In fact, if the model is taking LAI as an input rather than predicting it from some sort of allocation/turnover model that is downstream from the moss gas exchange model, then this needs to be much more dlearly delineated earlier on. Further, as you have discussed, TRIFFID is a dynamics vegetation model and as such predicts the distribution of moss as well as its physiology. Up to now, that element of model testing has been absent and therefore, I suspect, the scope of this model development needs greater elucidation in the methods and introduction sections.

P19 5-7: Also, I am not sure why LAI is described as a 'curve fitting' factor, when it has a clear biological meaning. Were no observations of LAI or FAPAR available? Is that a challenging aspect of working with moss?

P19 :L9-11: Is 1-3 the realistic rang eof LAI? Could GPP in principle be higher for optimal boundary conditions? I am not sure what to take from this description of the range.

P19 L 16: The 'Thus' in this sentence is redundant as it doesn't really follow from the discussion in the previous paragraph. Further, before making this statement, you need to reference the relevant figure and analysis. Also, the discussion of the 'residuals' is incomplete. Residuals of what? This seems to me to be an extremely abbreviated description of the results, and I am not sure, given that, that it is necessary to have these appendices.

Figure 2; Here it would have been helpful to illustrate what the default TRIFFID model does in the absence of the moss physiology. How wrong does it get the shape of these curves?

P21 L3: Here, again 'PFT model' is uninformative. Further, at this point, it would also be good to actually escribe the scope of the model that you have developed, which I

would describe as a 'gas exchange' model for non-vasculars plants.

P21 L6: On p19 L20, you state that you did in fact carry out 'calibration' and then here it is stated that no calibration was needed, before then in the next sentence again stating that Rg was calibrated, and then LAI? This obviously needs to be less contradictory. Also I think this is the first time you have discussed the literature on LAI in sphagnum?

P21 L12: There ARE insufficient test data. …

P22 L2: Simple climate correlation of what against what?

P22 L25: Many existing models have variable tissue Nitrogen concentrations, (see Davies Barnard et al. 2020) and other optimally adjust Vcmax as a function of environmental conditions. (see review in Franklin et al. 2020). In general,those types of model would be better equipped to deal with this issue (which is not particularly specific to mosses…) Again, the literature here needs to expand outside the JULES-TRIFFID domain.

P23 L9: The resolution of this issue, being the central development of the paper, does really need a better explanation in the main bulk of the text. Having read to here, but not the appendices, I am non-the-wiser about what was accomplish, nor why it was needed.

P23 L21: A DGCM does really imply that you are predicting the distribution of the individual PFTs. Have you tested that aspect of TRIFFID with spagnum, and if not, how did you test the model in the absence of competition with other vegetation types? The developments necessary to go from the gas exchange elements to a full biogeochemical and then dynamic vegetation scheme need to be at lest briefly discussed somewhere.

P32 L9: In my opinion you shouldn't be in any way grateful to the Hadley Centre for only allowing a part of this development to be made public. Continuing to retain the JULES code as a closed repository increasingly contravenes journal guidelines. I have, on many occasions as an editor and reviewer, been forced to make exceptions to journal

policies (to allow early career researchers to publish their hard work) because of this situation. For example, to get the JULES code here, I would need to break my reviewer anonymity and sign up for access, which violates the GMD policy as per:

"Where the authors cannot, for reasons beyond their control, publicly archive part or all of the code and data associated with a paper, they must clearly state the restrictions. They must also provide confidential access to the code and data for the editor and reviewers in order to enable peer review. The arrangements for this access must not compromise the anonymity of the reviewers. All manuscripts which do not make code and data available at this level are to be rejected. Where only part of the code or data is subject to these restrictions, the remaining code and/or data must still be publicly archived. In particular, authors must make every endeavour to publish any code whose development is described in the manuscript."

I appreciate that the author of this paper does not have jurisdiction to change this situation and has in fact gone to appropriate lengths to make the relevant code available under trying circumstances, but I find this feature of putting the onus on PhD students and early career scientist to force exceptions to journal rules to be highly regrettable.

References

Davies-Barnard, T., Meyerholt, J., Zaehle, S., Friedlingstein, P., Brovkin, V., Fan, Y., Fisher, R.A., Jones, C.D., Lee, H., Peano, D. and Smith, B., 2020. Nitrogen cycling in CMIP6 land surface models: Progress and limitations. Biogeosciences Discussions, pp.1-32.

Eller, C.B., Rowland, L., Mencuccini, M., Rosas, T., Williams, K., Harper, A., Medlyn, B.E., Wagner, Y., Klein, T., Teodoro, G.S. and Oliveira, R.S., 2020. Stomatal optimization based on xylem hydraulics (SOX) improves land surface model simulation of vegetation responses to climate. New Phytologist, 226(6), pp.1622-1637.

Franklin, O., Harrison, S.P., Dewar, R., Farrior, C.E., Brännström, Å., Dieckmann, U.,

Pietsch, S., Falster, D., Cramer, W., Loreau, M. and Wang, H., 2020. Organizing principles for vegetation dynamics. Nature plants, pp.1-10.

Walker, A.P., Johnson, A.L., Rogers, A., Anderson, J., Bridges, R.A., Fisher, R.A., Lu, D., Ricciuto, D.M., Serbin, S.P. and Ye, M. (2020), Multi‐hypothesis comparison of Farquhar and Collatz photosynthesis models reveals the unexpected influence of empirical assumptions at leaf and global scales. Global Change Biology. Accepted Author Manuscript. doi:10.1111/gcb.15366